# COVID-19 fear and its associated correlates among type-2 diabetes patients in Bangladesh: A hospital-based study†

Suvasish Das Shuvo[1]* , Md. Toufik Hossen[1]*, Md. Sakhawot Hossain[1] ,
Asma Khatun[1], Sanaullah Mazumdar[1] , Md. Riazuddin[1] and Deepa Roy[2]

[1]Department of Nutrition and Food Technology, Jashore University of Science and Technology, Jashore, Bangladesh and
[2]Department of Mathematics, Jashore University of Science and Technology, Jashore, Bangladesh

## Research Article

**Keywords:**
Bangladesh; COVID-19 fear; comorbidity; type-2 diabetes; predictors

**Corresponding author:**
Suvasish Das Shuvo;
Email: shuvo_nft@just.edu.bd

*S.D.S. and M.T.H. authors are equal contributions to this work.
†The online version of this article has been updated since original publication. A notice detailing the change has also been published

## Abstract

The outbreak of COVID-19 has caused widespread fear among people around the world, particularly those with underlying health conditions such as type-2 diabetes. This study aimed to investigate COVID-19 fear and its associated potential factors among type-2 diabetes patients in Bangladesh. A total of 1,036 type-2 diabetes patients residing in the Jashore district of Bangladesh were interviewed using the COVID-19 Fear Scale in Bengali language. A pre-validated questionnaire was used to collect data on sociodemographic, lifestyle-related characteristics, and COVID-19-related information. Logistic regression was performed to identify factors associated with perceived fear of COVID-19. The mean score of the COVID-19 fear was 18.1 ± 5.6. Approximately 45 and 39% were most afraid and uncomfortable thinking about COVID-19, respectively. Regression analysis revealed that gender, age, occupation, residence, physical activity, smoking, and dietary diversity score were associated with fear. Additionally, respondents who had limited self-care practice, unaffordable medicine, medicine shortages, a close friend or family member diagnosed with COVID-19, and financial problems during COVID-19 were significant predictors of COVID-19 fear. Healthcare providers should implement interventions, including appropriate education and counseling, to address the psychological impact of the COVID-19 pandemic on type-2 diabetes patients in Bangladesh.

## Impact statement

Fear arises as a fundamental emotional response to the perilous coronavirus disease, potentially serving as a contributing element in the development of diverse psychological disorders. The study illuminates the psychological obstacles experienced by type-2 diabetes patients by analyzing the extent of fear associated with COVID-19. This study's findings are essential for healthcare professionals and policymakers to comprehend the specific requirements and concerns of type-2 diabetes patients during the pandemic. The findings of this study not only expand our understanding of the psychological effects of COVID-19 but also offer evidence-based suggestions for customized interventions and support systems to reduce fear and enhance the well-being of individuals living with type-2 diabetes in Bangladesh. This research holds promise in guiding public health strategies and enhancing the overall management and care of diabetes patients in times of crisis, ensuring their safety and improved health outcomes.



## Introduction

The COVID-19 pandemic is the unprecedented new coronavirus and is considered one of the greatest public health issues globally in this century (Yan et al., 2020). Presently, people who have underlying chronic noncommunicable illnesses are more likely to be predisposed to the physical problems and mortality caused by severe acute respiratory syndrome coronavirus (Musche et al., 2021). In addition, COVID-19 infection has become a double challenge for people with diabetes and is the second most prevalent comorbidity (Yan et al., 2020). According to numerous research, the prevalence of diabetes in individuals with mild COVID-19 may range from 5.7 to 5.9% (Boschiero et al., 2021) whereas, the incidence of diabetes mellitus in patients with severe COVID-19 has increased dramatically from 22.2 to 26.9% (Kaplan Serin and Bülbüloğlu, 2021). Recent research reported that the death rate of patients with diabetes was three times higher than that of COVID-19 in patients as a whole (Jelinek et al., 2017; Seiglie et al., 2020). A previous study showed that more than half of the COVID-19 patients (52.4%) who were admitted to ICUs in Bangladesh had diabetes (Bäuerle et al., 2020).

In the context of significant mental diseases in low- and middle-income countries (LMICs), the direct and indirect impacts of COVID-19 on physical, psychological and social well-being are

already apparent (Sayeed et al., 2020; Ali and Elliott, 2021). However, diabetes and mental health disorders have become a synergistic epidemic, with biological, social and mental aspects all having an impact on overall health outcomes (Rubin and Peyrot, 2001). The risk of fear and anxiety is presumably higher in the elderly and those with comorbidities, who have higher hospitalization, illness severity, and mortality rates if infected with COVID-19 (Akhtar et al., 2020; Musche et al., 2021). Diabetes-related health issues with mental effects are greatly impacted by the fear of infection which increases the body's vulnerability to creating illness (Sayeed et al., 2020; Ali and Elliott, 2021). A previous study analyzed that 81.1% of people with diabetes were worried or feared because of the higher risk of COVID-19 infection in Bangladesh (Nataraj, 2021).

Due to the rapidly increasing demographic and epidemiological transition, Bangladesh is facing the challenge of health-related complications like diabetes (Akhtar et al., 2020). Among patients with diabetes, psychological distress and socioeconomic status (SES) are greatly influenced by a fear of COVID-19 (Alawadi et al., 2020; Al-Sofiani et al., 2021; Minoura et al., 2022). The primary reason is that diabetes patients experience fear as a result of their fragile immune systems, which would contribute to making their bodies more susceptible to infection during the COVID-19 pandemic (Jelinek et al., 2017; Nataraj, 2021). However, the invention of vaccine may change the pandemic waves but some groups of chronic disease patients feel fear because vaccinated people have also been infected again during that pandemic situation (Elsayed et al., 2022). Moreover, other important factors that increase fear among diabetic patients include the spread of fake news about the pandemic, ineffective lockdown, a lack of knowledge, delays in vaccination, and disruption of access to diabetes care services in Bangladesh (Al-Sofiani et al., 2021; Boschiero et al., 2021). As a result life-threatening consequences for people with diabetes due to greater fear of adverse outcomes from COVID-19 (Seiglie et al., 2020; Kaplan Serin and Bülbüloğlu, 2021). During the pandemic, a correlation has been observed between glycemic levels and psychological complications such as depression, distress, and anxiety (Bäuerle et al., 2020; Upsher et al., 2022). COVID-19-related fear also greatly influenced impaired Health-Related Quality of Life (HRQoL) and increased susceptibility to the risk of morbidity and mortality in diabetic patients (Erener, 2020; Puig-Domingo et al., 2020). Importantly, in Bangladesh, health expenditures for rapidly growing among diabetic patients are estimated at US$1.5 billion annually (Sridhar, 2016), which increases their financial burden and risk of severe health outcomes (Kang et al., 2021).

Considering this contextual concern, most studies in Bangladesh have focused on COVID-19-related fear among older adults (Islam et al., 2021; Mistry et al., 2021; Sujan et al., 2021) and only COVID-19-specific diabetes worries (Nataraj, 2021), but to our best knowledge, no studies have been undertaken to COVID-19 related fear and associated factors among type-2 diabetes patients during the pandemic. Therefore, this study was conducted to investigate the COVID-19 fear prevalence and its associated correlates among type-2 diabetes patients which will help to assist in implementing appropriate strategies, interventional as well as support programs in a low-resource setting like Bangladesh.

## Methods

### Study design and setting

A cross-sectional study was conducted in two diabetic hospitals (Ahad Diabetic and Health Complex, and *Kapotakkho Lions* Eye and Diabetic Hospital) residing in the Jashore district of Bangladesh from February to March 2022 during the third wave of the COVID-19 pandemic. To be eligible, the respondents had to be adults (≥21 years), Bangladeshi type-2 diabetes patients, currently living in Bangladesh at the time of the COVID-19 outbreak, and able to read and understand the Bengali language. Diabetes was defined according to the World Health Organization (WHO) criteria by the diabetic hospital's physician: Fasting plasma glucose (FPG) ≥7.0 mmol/L (126 mg/dL) (Safieddine et al., 2021). We excluded patients with type-1 diabetes, gestational diabetes, those on insulin therapy, and seriously ill patients not able to provide the interviews.

### Sample selection and data collection

This study was conducted with a total sample of 1,036 type-2 diabetes patients. The sample size was calculated using a 95% confidence level, 5% margin of error, 90% test power, and 95% response rate assuming 50% prevalence. We assumed that the psychological difficulties might be 50% among the type-2 diabetes patients of Bangladesh and so the calculated sample size was 384 participants. A simple random sampling technique was used to collect the data through face-to-face interviews using a structured questionnaire. The English version of the questionnaire was first translated into Bengali language and then back-translated into English by two bilingual experts to ensure content consistency. We pre-tested the questionnaire among 50 participants and after necessary modification, it was used for this study. Eight research assistants with experience in conducting health surveys were recruited for the data collection process. Before the data collection, the research assistants participated in Zoom meetings for in-depth training. Participants were informed about the study's goal, confidentiality, the opportunity to refuse participation, and their obligations. Before collecting data, all participants provided written informed consent.

### Outcome variable

COVID-19 fear was the primary outcome variable and the level of COVID-19-related fear was measured using the COVID-19 Fear Scale (FCV-19S) and as previously identified is a reliable and valid tool with seven items using a five-point Likert-scale ranging from 1 = "strongly disagree," 2 = "disagree," 3 = "neither disagree nor agree," 4 = "agree", and 5 = "strongly agree." The score ranged from 7 to 35, and the higher the score the greater the individual's fear of COVID-19 (Ahuja et al., 2021).

### Explanatory variables

Explanatory variables considered in this study were sociodemographic information (gender, age, education, occupation, monthly income, and residence), lifestyle-related characteristics (physical activity, physical exercise, smoking habit, duration of diabetes, and dietary diversity score [DDS]), anthropometric measurement of weight and height (Asian-specific body mass index [BMI] cutoffs) (Pan and Yeh, 2008), and challenges of getting routine medical and healthcare characteristics (transport difficulties, limited self-care practice, delayed care seeking, unaffordable medicine, medicine shortage, staff shortage, decreased inpatient capacity, close friend or family member diagnosed with COVID-19, financial problem during COVID-19) were calculated.

The survey included inquiries regarding participants' physical activity levels using the following question: "How often do you

participate in moderate or intense physical activity for a minimum of 30 min?" In this study, the gathered data was analyzed to evaluate the respondents' moderate to vigorous physical activity (MVPA), referencing a previous investigation (Perales et al., 2014). Subsequently, the responses were categorized into two levels: those meeting the recommended level of physical activity (more than three times a week, including daily engagement) and those falling below the recommended level (no physical activity, less than once a week, one or two times a week, and three times a week). The respondents' DDS were calculated based on information obtained from the 24-h dietary recall (Kennedy et al., 2010). To determine the DDS, 12 distinct food groups were taken into account, and each group's consumption during the specified period was assigned one point. If individuals reported consuming food from all 12 groups during the reference period, they received a maximum DDS of 12 points. The overall DDS ranges from 0 to 12 and is categorized into three groups: low (scores of 0–3), moderate (scores of 4–6) and high (scores of 7–12) (Kennedy et al., 2010). Self-reported information on pre-existing medical conditions, such as hypertension, hyperlipidemia, coronary artery disease, cerebrovascular disease, cardiovascular problem, kidney disease, asthma/COPD (chronic obstructive pulmonary disease) and arthritis were collected to identify comorbidity (Ke-You and Da-Wei, 2001; Akter et al., 2014). This information was verified by the participants, and subsequently, prescriptions were reviewed by a registered physician for the validity of the diagnosis. Participants with a history of hypertension, current use of anti-hypertensive medication, or a high blood pressure level during the interview were all considered. Hypertension diagnosis was based on a systolic blood pressure (SBP) reading of 140 mmHg and a diastolic blood pressure (DBP) reading of 90 mmHg (Mitra and Associates Dhaka, ICF, 2011).

### Statistical analysis

Descriptive statistics were employed to assess the distribution of variables. Mean differences in the FCV-19S score across participant characteristics were evaluated using independent *t*-tests and ANOVA. A binary logistic regression model was utilized to identify factors associated with fear, and the results of the regression analysis are reported as odds ratios (OR) with corresponding 95% confidence intervals (95% CI). All statistical analyses were conducted using Stata (Version 14.0) software package.

### Results

#### Participant's characteristics

Table 1 describes the sociodemographic and healthcare access characteristics of 1,036 type-2 diabetes patients. As shown in Table 1, most of the respondents were females (63.5%), aged between 50 and 64 years (44.5%), completed primary education (31.2%), manual workers (65.1%), residing in urban areas (59.9%) and do not undertake the recommended level of MVPA (28.3%). Among the study participants, 39.48% had a family history of diabetes, 42.4% had overweight, 46% had moderate DDS, and 23.3% had low DDS, respectively. Nearly, 13% had an anxiety disorder, and 41.4% had comorbid conditions. In addition, 19.2% of participants received delayed care seeking, 36.5% could not afford medication, 22.4% had medicine storage, 21.1% reported decreased inpatient capacity and nearly half of the participants

**Table 1.** Distribution of COVID-19 fear according to sociodemographic and healthcare access characteristics

| Variables | Category | Total *n* (%) | COVID-19 fear score Mean | SD | *p*-value |
|---|---|---|---|---|---|
| Total | | 1,036 (100) | 18.1 | 5.6 | |
| Sex | Female | 658(63.5) | 18.1 | 5.6 | 0.7906 |
| | Male | 378 (36.5) | 17.6 | 5.7 | |
| Age | Below 35 years | 90 (8.6) | 18.2 | 5.5 | 0.038 |
| | 35–49 years | 318 (30.7) | 17.5 | 5.5 | |
| | 50–64 years | 461 (44.5) | 18.0 | 5.6 | |
| | Above 65 years | 167 (16.2) | 18.9 | 5.9 | |
| Education | Graduates | 101 (9.7) | 17.6 | 5.2 | 0.004 |
| | HSC | 87 (8.4) | 17.5 | 4.5 | |
| | Secondary | 293 (28.3) | 17.4 | 5.4 | |
| | Primary | 323 (31.2) | 18.0 | 5.7 | |
| | Illiterate | 232 (22.4) | 19.2 | 6.1 | |
| Occupation | Manual worker | 676 (65.1) | 16.8 | 5.6 | 0.016 |
| | Nonmanual worker | 168 (9.2) | 18.5 | 5.7 | |
| | Unemployment/ retired | 192 (18.5) | 18.2 | 5.4 | |
| Monthly income | >20,000 BDT | 213 (20.6) | 17.9 | 5.8 | 0.748 |
| | 15,000–20,000 BDT | 330 (31.8) | 18.1 | 5.2 | |
| | 10,001–15,000 BDT | 318 (30.7) | 18.3 | 6.0 | |
| | <10,000 BDT | 140 (13.5) | 17.5 | 5.3 | |
| | Depend on other | 35 (3.4) | 17.8 | 6.5 | |
| Residence | Rural | 416 (40.1) | 17.7 | 4.8 (44.3) | 0.002 |
| | Urban | 620 (59.9) | 19.5 | 5.2 (60.9) | |
| Family history of diabetes | No | 627 (60.5) | 18.0 | 5.7 | 0.945 |
| | Yes | 409 (39.5) | 18.0 | 5.4 | |
| Physical exercise | Low (<30 min) | 349 (33.7) | 17.9 | 5.4 | 0.765 |
| | High (≥30 min) | 687 (66.3) | 18.1 | 5.7 | |
| MVPA | Less than the recommended level | 295 (28.3) | 19.6 | 5.6 | <0.001 |
| | Recommended level | 741 (71.7) | 17.4 | 5.5 | |
| Smoking habit | Nonsmoker | 852 (82.2) | 17.9 | 5.6 | 0.211 |
| | Ex-smoker | 114 (11.0) | 17.9 | 6.0 | |
| | Current smoker | 70 (7.8) | 19.2 | 5.3 | |
| BMI | Healthy weight | 261 (25.3) | 18.3 | 5.8 | 0.358 |
| | Underweight | 38 (3.6) | 17.1 | 5.8 | |
| | Overweight | 439 (42.4) | 17.7 | 5.4 | |
| | Obese | 298 (28.7) | 18.3 | 5.8 | |

(Continued)

**Table 1.** (*Continued*)

| Variables | Category | Total *n* (%) | COVID-19 fear score Mean | SD | *p*-value |
|---|---|---|---|---|---|
| DDS | High | 318 (30.7) | 18.3 | 5.7 | 0.218 |
|  | Moderate | 477 (46.0) | 18.0 | 5.8 |  |
|  | Low | 241 (23.3) | 17.5 | 5.0 |  |
| Anxiety | No | 905 (87.4) | 17.5 | 5.3 | <0.001 |
|  | Yes | 131 (12.6) | 21.4 | 6.4 |  |
| Comorbidity | No | 607 (58.6) | 17.4 | 5.7 | <0.001 |
|  | Yes | 429 (41.4) | 18.7 | 5.4 |  |
| Read or listened to news about the COVID-19 | No | 503 (48.5) | 17.1 | 5.3 | 0.002 |
|  | Sometimes | 370 (35.7) | 19.1 | 5.6 |  |
|  | Always | 163 (15.8) | 18.4 | 5.7 |  |
| Transport difficulties | No | 886 (85.5) | 18.0 | 5.6 | 0.675 |
|  | Yes | 150 (14.5) | 18.0 | 5.7 |  |
| Limited self-care practice | No | 908 (87.6) | 17.7 | 5.6 | 0.279 |
|  | Yes | 128 (12.4) | 19.8 | 5.2 |  |
| Delayed care seeking | No | 837 (80.8) | 17.7 | 5.5 | 0.552 |
|  | Yes | 199 (19.2) | 19.1 | 5.7 |  |
| Unaffordable medicine | No | 658 (63.5) | 18.0 | 5.9 | 0.002 |
|  | Yes | 378 (36.5) | 17.9 | 5.1 |  |
| Medicine shortage | No | 804 (77.6) | 18.0 | 5.6 | 0.680 |
|  | Yes | 232 (22.4) | 18.1 | 5.5 |  |
| Staff shortage | No | 937 (90.4) | 17.6 | 5.5 | 0.802 |
|  | Yes | 99 (9.6) | 21.1 | 5.4 |  |
| Decreased inpatient capacity | No | 817 (78.9) | 16.9 | 5.0 | <0.001 |
|  | Yes | 219 (21.1) | 21.8 | 6.1 |  |
| A close friend or family member diagnosed with COVID-19 | No | 887 (85.6) | 17.2 | 5.1 | 0.111 |
|  | Yes | 149 (14.4) | 23.0 | 5.7 |  |
| Financial problems during COVID-19 | No | 553 (53.4) | 16.9 | 5.3 | 0.016 |
|  | Yes | 483 (46.6) | 19.2 | 5.7 |  |

BDT, Bangladeshi taka; BMI, body mass index; DDS, dietary diversity score; MVPA, moderate to vigorous physical activity.

(46.6%) reported experiencing financial difficulties during COVID-19.

### COVID-19 fear among the participants

The mean score of the FCV-19S was 18.1 ± 5.6 among the respondents. Mean differences in the fear score of COVID-19 were noted by age, education, occupation, physical activity, smoking habit, anxiety, comorbidity, unaffordable medicine, decreased inpatient capacity, a financial problem during COVID-19, and a close friend or family member diagnosed with COVID-19 in Table 1. Participants reported agreement on the seven items of FCV-19S as shown in Figure 1. Approximately half of the participants (44.5%) were most afraid of COVID-19. More than one-third of the respondents (39%)

were uncomfortable thinking about COVID-19. Additionally, 20.3% of people have reported that their hands became clammy when they thought about COVID-19. When people viewed news and stories regarding COVID-19 on social media, 23.2% were experiencing anxiety or nervousness. However, 18.8% of respondents reported having trouble falling asleep because they worried about contracting COVID-19. Besides, 15% were afraid of losing their lives because of COVID-19.

### Factors associated with the COVID-19 fear

Table 2 presents the adjusted association between COVID-19 fear with demographic and healthcare characteristics. In the adjusted regression model, gender, age, occupation, residence, physical activity, smoking, DDS score, limited self-care practice, unaffordable medicine, medicine shortage, a close friend or family member diagnosed with COVID-19, and financial problem during COVID-19 were significantly associated with fear (FCV-19S). Females had approximately 4 times more fear of COVID-19 compared to males (OR = 3.83, 95% CI: 1.83–6.38), whereas ages between 50 and 64 years and above 65 years also showed 1.28 times and 1.51 times more fear than their counterparts (OR = 1.28, 95% CI: 1.12–2.46; OR: 1.51, 95% CI: 1.09–3.16). Regarding occupation, unemployed patients, and nonmanual workers were 2.47 times and 2.31 times (OR: 2.47, 95% CI: 1.76–4.17); OR: 2.31, 95% CI: 1.82–4.54) more fearful compared to the manual worker. It is also observed that patients residing in urban areas were 2.51 times (OR: 2.51, 95% CI: 1.24–4.16) more likely to fear compared with peers residing in rural areas. Moreover, type-2 diabetes patients undertaking a recommended level of MVPA (more than thrice to every day) had 0.66 times (OR: 0.66, 95% CI: 0.34–0.82) lower chances of being fear compared with peers performing less than the recommended level of physical activity. Additionally, the odds of being fear was 3.34 times (OR: 3.34, 95% CI: 1.42–5.32) and 1.21 times (OR: 1.21, 95% CI: 1.13–1.95) higher among current smoker and had low DDS, respectively, compared with their nonsmoker and high DDS counterparts. Again, those who had anxiety and comorbidity were 1.66 times and 1.43 times more likely to fear as compared to their counterparts (OR 1.66, CI: 1.27–3.53; OR 1.43, CI: 1.19–2.24). On the other hand, those who had limited self-care practice and unaffordable medicine were 3.49 times and 1.13 times higher odds (OR 3.49, CI: 1.27–5.76; OR 1.13, CI: 1.03–1.92) of being fear as compared to those peers. Lastly, we found type-2 diabetes patients who had medicine shortages, a close friend or family member diagnosed with COVID-19, and financial problems during COVID-19 were almost 2.27 times, 3.83 times, and 2.92 times higher risk of being fear as compared to their peers (OR: 2.27, CI: 1.24–4.16; OR: 3.83, CI: 1.42–6.35; OR: 2.92, CI: 1.54–4.58), respectively.

### Discussion

Individuals in Bangladesh with chronic conditions, such as type-2 diabetes, may experience increased fear and anxiety due to the elevated risk of severe illness or death from COVID-19. This research article aimed to explore the prevalence and associated factors of fear among type-2 diabetes patients in Bangladesh during the COVID-19 pandemic. In our study, approximately half of the participants (44.5%) were most afraid and uncomfortable thinking (39%) about COVID-19. Additionally, respondents not only experienced anxiety or nervousness (23.2%) but also had anxiety disorder and comorbid conditions about 13 and 41.4%, respectively. Similar results were seen in earlier various studies (Yan et al.,

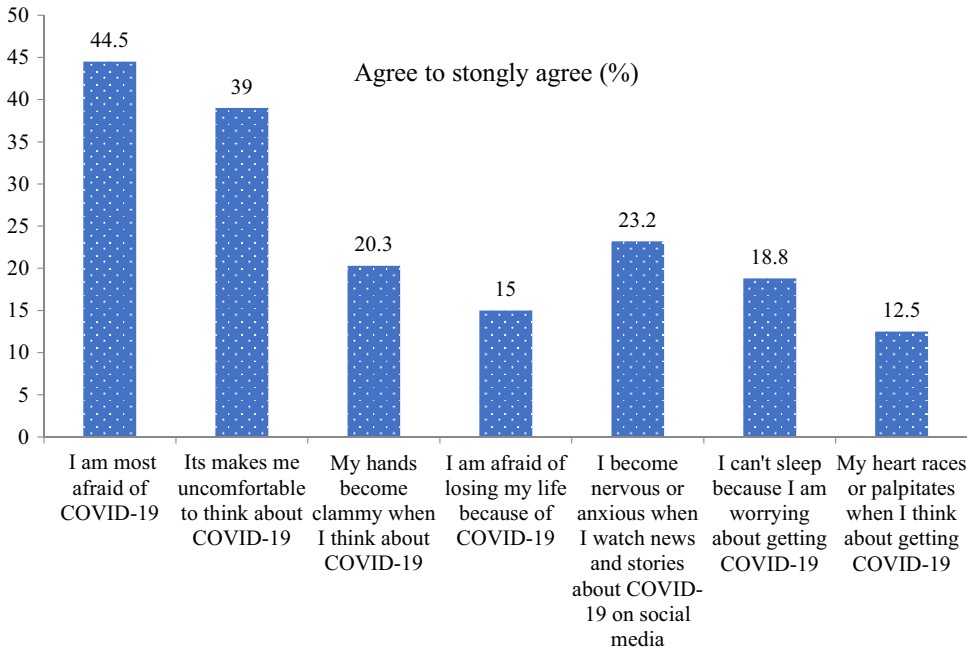

**Figure 1.** Participant's agreement on seven items of the COVID-19 Fear Scale (FCV-19S).

2020; Al-Rahimi et al., 2021; Musche et al., 2021; Sujan et al., 2021). This may be related to the perception of higher infection risk, quicker transmission, loneliness, lack of cure and vaccine, increased mortality rate, and uncertainty about the future (Ahuja et al., 2021; Al-Sofiani et al., 2021; Sujan et al., 2021). The present study revealed that females were more likely to experience fear than males during COVID-19. Many consistent findings were reported by researchers who conducted their research during the COVID-19 pandemic situation (Bakioğlu et al., 2021; Basit et al., 2021; Islam et al., 2021; Ayaz-Alkaya and Dülger, 2022). One possible explanation for this finding is that males may fall sick less frequently or refrain from expressing fear due to prevailing gender roles (Bakioğlu et al., 2021). Additionally, women tend to be more emotionally reactive and are more likely to perceive negative information, which could contribute to increased levels of COVID-19 fear (Hankin et al., 1998). Our recent study identified that diabetes patients aged between (50–64) years and above 65 years were more likely to feel fear than their counterparts. The study observation is consistent with previous studies conducted in Bangladesh and Switzerland (Mistry et al., 2021; Ganguli et al., 2022). This could be due to a combination of age-related physiological changes, vulnerabilities including unemployment, social isolation, and difficulty in accessing medications and health care is more susceptible to the negative effects of COVID-19 (Mistry et al., 2021; WHO, 2023). The findings of this study further revealed that diabetes patients with lower levels of education and unemployed or retired were significantly associated with higher fear than manual workers. Several similar studies were reported in Bangladesh, and Pakistan (Ayaz-Alkaya and Dülger, 2022; Bilal et al., 2022). It was regarded as a lack of knowledge about COVID-19, unfamiliarity with the virus, feelings of helplessness, and loneliness extended fear during the COVID-19 pandemic (Al-Rahimi et al., 2021; Mistry et al., 2021). Due to the lockdown and forced isolation during the pandemic, unemployed and retired individuals have experienced a decline in their ability to

communicate with others compared to pre-pandemic times. This could result in feelings of loneliness and isolation among them (Robb et al., 2020). Our study also observed that diabetes patients who reside in urban areas were more likely to have a fear of the COVID-19 virus than people who live in rural areas. Similar results were obtained in prior studies conducted in Bangladesh and China (Rahman et al., 2020a; Sujan et al., 2021). There may be more crowded people and greater access to media and health information leading to higher risks of COVID-19-related fear among urban diabetes patients (Huang et al., 2021; World Economic Forum., 2022). Our study also showed that diabetes people who engaged in regular MVPA in recommended level had decreased probability of fear related to COVID-19, compared to those who did not exercise. Compared to studies conducted in Dubai and Iran, where the reduced risk of fear during the COVID-19 pandemic condition (Ali and Elliott, 2021; Peimani et al., 2022). This could be regarded as they provide emotional support, meeting physical activity guidelines that were associated with COVID-19-specific diabetes patients (Peimani et al., 2022). The current investigation confirmed that type-2 diabetes patients who undertook a recommended level of MVPA were less likely to experience fear than those who did not undertake it at the time of COVID-19. During the COVID-19 pandemic, a more consistent study was conducted in Brazil (de Paiva Teixeira et al., 2020). There could be several reasons why this might be the case: Regular exercise and physical activity can help manage blood sugar levels and improve insulin sensitivity in people with type-2 diabetes (Colberg et al., 2016). Improved glycemic control may reduce the risk of developing severe COVID-19 complications, as high blood glucose levels have been associated with worse outcomes in COVID-19 patients (Pal and Bhadada, 2020). Our study also reported that during the COVID-19 pandemic diabetes patients with current smokers were more fearful than nonsmoker. Many similar studies have shown that the situation became tenser since diabetes patients did not maintain proper

**Table 2.** Association between COVID-19 fear with demographic and healthcare characteristics

| Variables | Category | OR (95% CI) | *p*-value |
|---|---|---|---|
| Gender | Male | 1 | |
| | Female | 3.83 (1.83–6.38) | 0.004 |
| Age | Below 35 years | 1 | |
| | 35–49 years | 0.77 (0.32–1.89) | 0.576 |
| | 50–64 years | 1.28 (1.12–2.46) | 0.005 |
| | Above 65 years | 1.51 (1.09–3.16) | 0.041 |
| Education | Graduates | 1 | |
| | HSC | 0.76 (0.25–2.27) | 0.559 |
| | Secondary | 0.43 (0.17–1.04) | 0.056 |
| | Primary | 1.50 (1.18–2.30) | 0.013 |
| | Illiterate | 1.78 (1.28–2.79) | 0.021 |
| Occupation | Manual worker | 1 | |
| | Nonmanual worker | 2.31 (1.82–4.54) | 0.005 |
| | Unemployment or retired | 2.47 (1.76–4.17) | 0.001 |
| Family monthly income | >20,000 BDT | 1 | |
| | 15,000–20,000 BDT | 1.39 (1.26–2.51) | 0.029 |
| | 10,001–15,000 BDT | 0.85 (0.46–1.53) | 0.552 |
| | <10,000 BDT | 0.95 (0.45–2.01) | 0.884 |
| | Depend on other | 0.26 (0.09–0.75) | 0.013 |
| Residence | Rural | 1 | |
| | Urban | 2.51 (1.24–4.16) | 0.021 |
| Family history of diabetes | No | 1 | |
| | Yes | 1.37 (0.85–2.13) | 0.22 |
| Physical exercise | No | 1 | |
| | Yes | 0.37 (0.17–0.76) | 0.039 |
| MVPA | Less than the recommended level | 1 | |
| | Recommended level | 0.66 (0.34–0.82) | 0.019 |
| Smoking habit | Nonsmoker | 1 | |
| | Ex-smoker | 0.33 (0.15–0.70) | 0.005 |
| | Current smoker | 3.34 (1.42–5.32) | 0.030 |
| BMI | Healthy weight | 1 | |
| | Underweight | 0.39 (0.13–1.10) | 0.079 |
| | Overweight | 0.70 (0.40–1.21) | 0.219 |
| | Obese | 0.61 (0.33–1.12) | 0.112 |
| DDS | High | 1 | |
| | Moderate | 0.72 (0.43–1.19) | 0.208 |
| | Low | 1.21 (1.13–1.95) | 0.001 |
| Anxiety | No | 1 | |
| | Yes | 1.66 (1.27–3.53) | 0.015 |
| Comorbidity | No | 1 | |
| | Yes | 1.43 (1.19–2.24) | 0.025 |

*(Continued)*

**Table 2.** (*Continued*)

| Variables | Category | OR (95% CI) | *p*-value |
|---|---|---|---|
| Read or listened to news about the COVID-19 | No | 1 | |
| | Sometimes | 3.15 (1.53–5.78) | 0.002 |
| | Always | 2.07 (1.13–3.06) | 0.024 |
| Transport difficulties | No | 1 | |
| | Yes | 1.52 (0.27–2.98) | 0.244 |
| Limited self-care practice | No | 1 | |
| | Yes | 3.49 (1.27–5.76) | 0.015 |
| Delayed care seeking | No | 1 | |
| | Yes | 0.93 (0.51–1.77) | 0.991 |
| Unaffordable medicine | No | 1 | |
| | Yes | 1.13 (1.03–1.92) | 0.038 |
| Medicine shortage | No | 1 | |
| | Yes | 2.27 (1.24–4.16) | 0.008 |
| Staff shortage | No | 1 | |
| | Yes | 1.31 (0.51–3.39) | 0.625 |
| Decreased inpatient capacity | No | 1 | |
| | Yes | 0.98 (0.46–2.11) | 0.968 |
| A close friend or family member diagnosed with COVID-19 | No | 1 | |
| | Yes | 3.83 (1.42–6.35) | 0.008 |
| Financial problems during COVID-19 | No | 1 | |
| | Yes | 2.92 (1.54–4.58) | 0.007 |

BDT, Bangladeshi taka; BMI, body mass index; DDS, dietary diversity score; MVPA, moderate to vigorous physical activity; OR, odds ratios

guidance during the pandemic situation (Basit et al., 2021; Reddy et al., 2021; Sujan et al., 2021). This could be because smoking can cause damage to the respiratory system, making smokers more vulnerable to respiratory infections such as COVID-19 (Nidhi Saha, BDS, 2022). Moreover, diabetes patients who smoke can weaken their immune system and are more likely to have other health conditions, such as heart disease and lung disease, which can also increase the risk of severe illness and complications if they contract COVID-19 (WHO, 2021a). Data from this study suggest that diabetes patients with lower DDS were more likely to have higher COVID-19-specific fear compared with high DDS counterparts. Accordingly, a study was conducted in Ethiopia among diabetes patients at this critical moment (Alenko et al., 2021). Because of poor socioeconomic status, inadequate nutritional diet, and strict lockdown measures were associated with poor consumption during the COVID-19 pandemic (Seiglie et al., 2020; Kim et al., 2022). Therefore, patients with diabetes who have poor dietary diversity may have weaker immune systems and making them more vulnerable to COVID-19 infection (Hussien et al., 2021). Our study findings further showed that diabetes patients who had anxiety and comorbidity were more likely to fear as compared to their counterparts. Equivalent studies were published in numerous countries (Jelinek et al., 2017; Alenko et al., 2021; Abdelghani et al., 2022). This could be due to individuals with chronic illness may experience heightened vulnerability to COVID-19, as well as potential challenges accessing medical treatment and follow-up care, potentially leading to feelings of depression and fear related to COVID-19

(Li et al., 2020). Also, we found that diabetes patients who occasionally or always read or listened to the news about COVID-19 experienced higher fear during this pandemic situation. Two more consistent studies were obtained in Bangladesh and Turkey (Rahman et al., 2020a; Ayaz-Alkaya and Dülger, 2022). The possible explanation is that media coverage of COVID-19 has been dominated by alarming statistics and sensationalized headlines, which can exacerbate anxiety and fear among viewers and listeners (Ali and Elliott, 2021). This constant barrage of negative news can cause individuals to perceive the situation as more dangerous and threatening than it is, leading to heightened anxiety and fear. Moreover, diabetes patients may be particularly vulnerable to these stressors due to their underlying health condition, which requires careful management and lifestyle adjustments. News coverage that emphasizes the severity of the pandemic can add to this stress and exacerbate feelings of fear and anxiety (Holmes et al., 2020). The current study identified that diabetes patients who had limited self-care practice were feeling highly fearful compared to their peers. The equivalent study was conducted in Germany for the estimation of self-care practices due to the impacts of COVID-19 (Fiske et al., 2021). This could be since of poorly controlled diabetes can lead to serious health complications, including blindness, kidney disease, heart disease, and neuropathy, among others (Ali and Elliott, 2021). Lack of healthcare practices, improper use of masks and hand sanitizer, and failure to maintain social distancing could all contribute to diabetes patients' elevated levels of fear (Kaplan Serin and Bülbüloğlu, 2021; WHO, 2021b). Our current study revealed that type-2 diabetes patients who had unaffordable medicine and medicine shortages during COVID-19 were more likely to be fearful compared to their peers. Similarly, previous research was established in China and Saudi Arabia (Yan et al., 2020; Al-Sofiani et al., 2021; Abdelghani et al., 2022). This is verified in recently published evidence which documented those older adults from LMICs faced various challenges, including access to medicine and routine medical care amid this COVID-19 pandemic (Ssewanyana et al., 2018). As a developing country, during lockdown and isolation many people from Bangladesh faced problems accessing their daily essentials (Shammi et al., 2021). For older adults, medicines are essential especially as they often suffer from different comorbidities and in cases multiple morbidities (Chow et al., 2018).

The unavailability of required medicines can have a serious impact on their mental health resulting in fear. A significant finding of our study is that respondents' close friends or family members diagnosed with COVID-19 were significantly identified as fearful. Two consistent studies were conducted in Bangladesh during this COVID-19 pandemic (Mistry et al., 2021; Quadros et al., 2021). This could be the fact that adults who suffered from pre-existing mental illness (Rahman et al., 2020b) and those whose friends or contacts were hospitalized with COVID-19. The present study also indicates that the COVID-19 pandemic created significant fear among the respondents who had financial problems were more fearful. A more consistent study was conducted in Bangladesh, among diabetes patients (Mistry et al., 2021). Amidst the COVID-19 lockdown in Bangladesh, numerous family members who were breadwinners lost their jobs, especially garment workers (UCA News, 2020). Additionally, the harvesting and transportation of crops and vegetables were delayed, leading to food shortages and price hikes (Heifer International, 2020). As a result, people's earnings are declining each day due to the disease outbreak, while food prices are skyrocketing, causing a rise in anxiety among the populace.

The findings of this study suggest that COVID-19 fear is prevalent among type-2 diabetes patients in Bangladesh, and several factors are associated with this fear, including age, gender, education, income, and comorbidities. Policymakers and community health workers should develop and implement targeted interventions to address the specific needs of type-2 diabetes patients during the pandemic. This could be including improving education programs, providing mental health support, information on infection prevention, and access to telemedicine. As a developing country, the government should need to enhance health communication strategies and free diabetic campaign services among those underprivileged diabetes patients for diminished pandemic-specific fear. However, they also need to raise public awareness of the pandemic scenario among the most vulnerable diabetes patients, encouraging them to wear masks, keep social distancing, and practice good hand hygiene. Further research should be promoted to better understand the impact of COVID-19 on type-2 diabetes patients and to develop effective strategies to mitigate the negative effects of the pandemic on their health and well-being. Finally, this study highlights the importance of developing culturally appropriate interventions that consider the unique needs and perspectives of diabetes patients in Bangladesh.

### Strengths and limitations

This research addresses an important and timely topic with significant public health implications. The study focuses on a specific population, type-2 diabetes patients in Bangladesh, which allows for a detailed analysis of the impact of COVID-19 fear on this vulnerable group. We addressed participants by using a face-to-face interview-based survey by taking adequate protections. Moreover, the study uses a hospital-based cross-sectional design with a large sample size, which provides a relatively high level of confidence in the results obtained. We used validated and reliable tools for data collection, which increases the validity and reliability of the findings. Appropriate statistical methods were used to analyze the data and draw conclusions. Our study has not gone beyond some limitations. The study relies on self-reported data, which may be subject to recall bias and social desirability biases. The cross-sectional design limits the ability to draw causal inferences. The study focuses on a specific population in a specific context, which limits the generalizability of the findings to other populations and contexts. Additionally, because type-2 diabetes was the only condition considered, its results might not apply to other groups.

### Conclusion

This study shows that during the COVID-19 pandemic, patients with type-2 diabetes had a greater prevalence rate of fear. Most patients also reported being afraid and concerned about their own and their family health during the COVID-19 pandemic. Age, financial hardship, lack of access to healthcare, and lack of understanding about the pandemic were the main contributing factors to fear among them. However, this study emphasizes how critical it is to offer sufficient assistance to type-2 diabetes patients in Bangladesh during the epidemic, as well as the necessity of raising awareness, improving access to healthcare, and strengthening financial resources. Additionally, the study findings suggest clear evidence that healthcare providers offer diabetes patients support and helplines throughout and after the COVID-19 pandemic. Finally, it is intended that by implementing these procedures, the patient's fear and related factors will be lessened, improving their physical and mental health results.

## Abbreviations

| | |
|---|---|
| BDT | Bangladeshi taka |
| BMI | body mass index |
| DDS | dietary diversity score |
| FCV-19S | COVID-19 Fear Scale |
| MVPA | moderate to vigorous physical activity |
| OR | odds ratios |
| WHO | World Health Organization |

**Open peer review.** To view the open peer review materials for this article, please visit http://doi.org/10.1017/gmh.2023.47.

**Data availability statement.** There are some restrictions on this data, and it is not available to the public. Data will be available upon reasonable request.

**Acknowledgments.** All the authors wish to express their gratitude to the participants who volunteered for this study.

**Author contribution.** S.D.S. and M.T.H. conceptualized the study, synthesized the analysis plan and conducted the statistical analysis. S.D.S., M.T.H. and M.S.H. compiled the data and interpreted the findings. M.T.H., S.D.S., M.S.H., A.K., S.M., M.R. and D.R. drafted the manuscript, critically reviewed the manuscript and approved the manuscript. S.D.S. and M.T.H. have made equal contributions to this work.

**Financial support.** This research received no specific grant from any funding agency in the public, commercial or not-for-profit sectors.

**Competing interest.** There is no conflict of interest among the authors.

**Ethics statement.** This study was conducted according to the guidelines laid down in the Declaration of Helsinki and all procedures involving research study participants were approved by the Ethical Review Committee of the Faculty of Biological Science and Technology, Jashore University of Science and Technology (Ref: ERC/FBST/JUST/2022-107). Written and verbal informed consent was obtained from all patients. All information provided by respondents was ensured to remain confidential. The respondents were informed that they could withdraw at any time throughout the interview.

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
