## [Reviewer Report]

6/6/2023

The Editor

Global Mental Health 

Dear Sir, 

It is with pleasure that we are submitting our manuscript entitled “COVID-19 Fear and its Associated Predictors among Type-2 Diabetes Patients in Bangladesh: A hospital-based study” for consideration to Global Mental Health. This manuscript provides original work which is not under consideration for publication elsewhere. All authors approved the manuscript and this submission. 

COVID-19 has become a global pandemic affecting millions of people worldwide. Type-2 diabetes patients are considered to be at high risk of severe complications if infected with COVID-19. In Bangladesh, where the prevalence of diabetes is increasing, the fear of COVID-19 among diabetic patients may have a significant impact on their mental health and quality of life. Therefore, we conducted a hospital-based study to investigate the level of COVID-19 fear and its associated predictors among type-2 diabetes patients in Bangladesh. We found that a significant proportion of participants had a high level of fear of COVID-19, which was associated with various sociodemographic and clinical factors. 

We believe that the findings of our study will provide valuable insights into the psychological impact of COVID-19 on type-2 diabetes patients in Bangladesh. We also hope that our study will contribute to the existing literature on COVID-19 fear and its associated predictors among diabetic patients, particularly in low- and middle-income countries. 

We author feel “Global Mental Health” is the perfect scientific platform to disseminate the findings all over the world. We believe that our study meets the scope and criteria of your journal and will be of interest to your readers. We look forward to your favorable consideration of our manuscript for publication in your journal.

Thank you for your consideration! 

Regards, 

Suvasish Das Shuvo

(The corresponding author)

---

## [Reviewer Report]

The study titled “COVID-19 Fear and its Associated Predictors among Type-2 Diabetes Patients in Bangladesh: A hospital-based study” is an interesting and important scientific evidence. Overall, it is a well written and well-presented manuscript with appropriate statistical analysis. However, few comments to include and a minor revision are needed before publication:

Title

Consider revising it to reflect the key variables studied and the population of interest, such as " COVID-19 Fear and its Associated Correlates among Type-2 Diabetes Patients in Bangladesh: A hospital-based study

Abstract

Page 1, line 5-7: Please correct it accordingly “This study aimed to investigate COVID-19 fear and its associated potential factors among type-2 diabetes patients in Bangladesh.”

Introduction

The introduction part provides clear background about COVID-19 Fear and its Associated Predictors among Type-2 Diabetes Patients in Bangladesh and the objective of the study is quite clear.

Page 1, line 32: Correct reference style in text.

Page 2, line 51-52: Correct reference style in text.

Page 2, line 66-68: need to rewrite in quite simpler form

Page 3, line 70: “weak lockdown” ?? Is it correct?

Page 3, line 74-75: fail to well connect with the ongoing discussion

Page 3, line 78-81: fail to well connect. Need to reorganize this sentence

Methods:

Page 3, line 91: Write “a total of” instead of a group of

Page 3, line 91-93: Clarify the specific diabetic hospitals from which the participants were recruited in the Jashore districts of Bangladesh.

Page 3-4, line 100-103: Rewrite these sentence

Page 4, line 107-109: Consider Rewrite these sentences

Page 4, line 119: Shortly describe how you measure Moderate to vigorous physical activity (MVPA), dietary diversity score, body mass index (BMI) with references in method section.

Page 5, line 36-37: Rewrite this sentence

Results

Page 5, line 161-162: Write COVID-19 Fear instead Fear in Table 1: Distribution of Fear according to socio-demographic and healthcare access characteristics

Page 6, line 181: Write COVID-19 Fear instead of fear of COVID-19

Page 6, line 197: need to focus clearly on physical activity

Discussion

Page8, line 232: need to write year (Hankin et al., n.d.)

Page8, line 238-239: need to write year in ref. (Mental Health and Well-Being, n.d.)

Page8, line 250-251: need to rewrite

Page8, line 254: need to correct text ref. (How to Make Living in Cities Healthier for Everyone, According to an Expert, 2022;

Page8, line 255-256: physical activity?? Is it write in this sentence please correct it.

Page9, line 278-279: need to correct text ref. (Global Diabetes Summit, n.d.; What You Need to Know: Treatments for COVID-19 | ADA, n.d.)

Page10, line 311: need to correct text ref. (When and How to Use Masks, n.d)

Page10, line 332-333: need to correct text ref. (Covid-19, Job Cuts and Misery for Bangladesh’s Garment Workers - UCA News, n.d.)

Page10, line 335-336: need to correct text ref. (In Bangladesh COVID-19 Puts Farmers and Food Systems in Dire Straights, n.d.)

Page11, line 344: Write full word of “govt.”

I think authors should segregate policy implication and Strengths and limitations in two paragraph.

The article could benefit from a more thorough proofreading to improve the clarity and organization of the writing. Additionally, there are a few typographical errors and citation formatting throughout the text.

Overall comments:

The article provides a clear view of Covid-19 related fear and the associated facts that lead to severe risk among older adults. With some minor modifications and grammatical corrections, the article is well conducted with clear methodology and objective, and will provide a great help in public health interventions. I suggest accept for publication after minor revision.

---

## [Reviewer Report]

Keywords: The author should use Keywords in order from A to Z. Please re-write Covid to COVID.

Abstract: well organized and easy to understand for readers.

Introduction:

-Line 38 and 39: covid-19 should be written to COVID-19

-Problems with diabetes and COVID-19 were described repeatedly in the introduction section. It should be revised.

-Concise introduction will be better for the reader. It seems like a long introduction with the same statement.

Methods:

-Line 95: Sample size to sample size

-How can you assume the psychological difficulties?

-Study design should be short.

-How could you measure diabetes and hypertension?

Results:

-It would be better to compare fear about COVID-19 with normal peoples and CVDs patients

Discussions:

Last part of discussion should be concise.

References: well organized

---

## [Reviewer Report]

25/7/2023

The Editor

Global Mental Health 

Dear Sir, 

It is with pleasure that we are submitting our manuscript entitled “COVID-19 Fear and its Associated Correlates among Type-2 Diabetes Patients in Bangladesh: A hospital-based study” for consideration to Global Mental Health. This manuscript provides original work which is not under consideration for publication elsewhere. All authors approved the manuscript and this submission. 

COVID-19 has become a global pandemic affecting millions of people worldwide. Type-2 diabetes patients are considered to be at high risk of severe complications if infected with COVID-19. In Bangladesh, where the prevalence of diabetes is increasing, the fear of COVID-19 among diabetic patients may have a significant impact on their mental health and quality of life. Therefore, we conducted a hospital-based study to investigate the level of COVID-19 fear and its associated predictors among type-2 diabetes patients in Bangladesh. We found that a significant proportion of participants had a high level of fear of COVID-19, which was associated with various sociodemographic and clinical factors. 

We believe that the findings of our study will provide valuable insights into the psychological impact of COVID-19 on type-2 diabetes patients in Bangladesh. We also hope that our study will contribute to the existing literature on COVID-19 fear and its associated predictors among diabetic patients, particularly in low- and middle-income countries. 

We author feel “Global Mental Health” is the perfect scientific platform to disseminate the findings all over the world. We believe that our study meets the scope and criteria of your journal and will be of interest to your readers. We look forward to your favorable consideration of our manuscript for publication in your journal.

Thank you for your consideration! 

Regards, 

Suvasish Das Shuvo

(The corresponding author)